



# A Double Multiple Streamtube model for Vertical Axis Wind Turbines of arbitrary rotor loading

Anis A. Ayati[1,2], Konstantinos Steiros[1], Mark A. Miller[1], Subrahmanyam Duvvuri[1,3], and Marcus Hultmark[1]

[1]Department of Mechanical and Aerospace Engineering, Princeton University, Princeton, NJ 08544, USA
[2]Department of Mathematics, University of Oslo, Oslo, N-316, Norway
[3]Department of Aerospace Engineering, Indian Institute of Science, Bengaluru, 560012, India

**Correspondence:** K. Steiros (ksteiros@princeton.edu)

**Abstract.** We introduce an improved formulation of the Double Multiple Streamtube (DMST) model for the prediction of the flow quantities of Vertical Axis Wind Turbines (VAWT). The improvement of the new formulation lies in that it renders the DMST valid for any induction factor, i.e. for any combination of rotor solidity and tip speed ratio. This is done by replacing the Rankine-Froude momentum theory of the DMST, which is invalid for moderate and high induction factors, with a new momentum theory recently proposed, which provides sensible results for any induction factor. The predictions of the two DMST formulations are compared with VAWT power measurements obtained at Princeton's High Reynolds number Test Facility, over a range of tip speed ratios, rotor solidities and Reyonlds numbers, including those experienced by full scale turbines. The results show that the new DMST formulation demonstrates a better overall performance, compared to the conventional one, when the rotor loading is moderate or high.

## 1 Introduction

The study of vertical-axis wind turbines (VAWTs) has received renewed attention in the last decade. There were noticeable research efforts devoted to VAWTs from the mid 1970s to the mid 1980s, primarily led by Sandia National Laboratories and NASA (Strickland, 1975, 1987; Sheldahl and Klimas, 1981; Paraschivoiu et al., 1983; Paraschivoiu, 1981; Muraca et al., 1975). The following two decades saw relatively little research activity on the topic, as it was concluded that VAWTs were more prone to fatigue, and less efficient than horizontal-axis wind turbines (HAWTs). Recently, however, it was suggested that by tightly packing VAWT in a wind farm one can achieve increased power output per land area, compared to large modern HAWT farms (Dabiri, 2011). The above, coupled with the fact that VAWTs are insensitive to wind direction, have a low center of gravity, are serviceable from the ground and offer low manufacturing and maintenance costs, have created a resurgence of interest in VAWT wind farms.

An important prerequisite for the successful realization of wind farms is the development of engineering flow models that combine low computational cost and sufficient accuracy, so that they can be used as design and optimization tools. In the case of HAWTs, Blade Element Momentum algorithms (BEM) have been shown to fulfill these conditions, and have subse-





quently become standard aerodynamic tools of the HAWT industry. A significant amount of research has been devoted to the development of analogous models for the case of VAWTs.

This is not a trivial matter, however, as the aerodynamics that govern VAWTs are inherently more complex than HAWTs. The effective angle of attack experienced by a VAWT blade section is not constant, as in the case of HAWTs, but depends on the blade's instantaneous orbital position as well as on the tip speed ratio (ratio of turbine tip to free stream velocities). In addition, at relatively low tip speed ratios a blade section may experience large and rapid variations in effective angle of attack over the course of one rotation cycle. This leads to the highly unsteady and non-linear flow phenomenon known as dynamic

stall (Simão Ferreira et al., 2009; Buchner et al., 2015, 2018), which causes significant hysteresis in drag and lift forces. Lastly, depending on the tip speed ratio and rotor solidity, a blade located on the downwind rotor section may interact with its own or another blade's wake generated upwind (Kozak et al., 2016; Posa and Balaras, 2018), complicating further the VAWT response.

Despite these inherent complexities, a number of simplified analytical predictive methodologies have been proposed over the years (e.g. vortex, cascade, fixed wake, streamtube approaches (Islam et al., 2008; Wilson and McKie, 1980)). The streamtube,

and specifically its variant, the Double Multiple Streamtube (DMST) model (Paraschivoiu, 1981; Rolin and Porté-Agel, 2018) is one of the most popular approaches, due to its low computational cost, relative robustness and easiness of implementation. In a DMST model, the flow domain is discretized into a set of adjacent streamtubes, each featuring two actuators in tandem, one in the upstream rotor half-cycle and the second in the downstream half-cycle. In that way, two important aspects of the flow physics are taken into account: the constantly changing flow conditions experienced by the blades, and the difference in

perceived wind between the front and rear part of the rotor.

Nevertheless, such treatment of the rotor fails to model other important aspects of the flow physics: DMST assumes zero expansion of the streamtubes, it neglects the wake-blade interaction and it overlooks the effect of the downstream half of the rotor on the upstream half. For these reasons, DMST algorithms are known to fail to accurately capture the local aerodynamic loads on the rotor (Wilson and McKie, 1980; Ferreira et al., 2014); still, their "global" or mean predictions are of significant

accuracy, and as a result DMST remains a popular tool in VAWT design protocols.

Despite its usefulness, however, DMST is inapplicable to highly loaded VAWTs, i.e. characterized by high values of rotor solidity and tip speed ratio. That is because rotor loading correlates with the induction factors of the streamtubes. At an induction factor of 50% the core of the DMST model, the "classical" momentum theory of Rankine-Froude breaks down, predicting zero wake velocity and infinite wake width. For even larger induction factors the wake velocities and wake widths

assume nonphysical negative values, while drag is greatly underpredicted (Hansen, 2015).

In HAWT BEM models, this inconsistency of the momentum theory is rectified by using empirical values for the drag, the so called "Glauert's correction" (Buhl and Marshall, 2005). In the case of VAWTs, however, this is not sufficient as the wake flow quantities need to be corrected as well. That is because in a DMST solution the wake velocity of the front half-rotor determines the response of the rear half-rotor. Classical momentum theory cannot accurately predict the wake flow quantities

at high induction factors. As such, DMST is typically considered valid only for weakly loaded rotors where the induction factor is smaller than 50% (Ferreira et al., 2014).





In this article, we propose a resolution to this issue by substituting the Rankine-Froude momentum theory of the DMST with the momentum theory proposed by Steiros and Hultmark (2018). This momentum theory takes into account the effect of "base suction" in the wake (i.e. low wake pressure due to dissipation and wake entrainment), which is neglected in the the theory of Rankine-Froude. For low induction factors, where base suction is minimal, the predictions of the two momentum theories coincide, while for large induction factors the theory of Steiros and Hultmark produces much more realistic predictions. In that way DMST becomes valid, in principle, for any rotor loading.

To quantify the accuracy of the proposed methodology, we compare predictions of a conventional DMST model equipped with both the momentum theories of Rankine-Froude and Steiros and Hultmark, and with VAWT data acquired at Princeton's High Reynolds number Test Facility (HRTF). The data cover a range of rotor solidities, Reynolds numbers and tip speed ratios, which involve both "weakly" and "heavily" loaded rotors, at dynamically similar conditions to field VAWTs.

Following this introduction, the most relevant steps of the DMST model are outlined in section 2. The HRTF experiments are briefly described in section 3, results are discussed in section 4, and concluding remarks are given in section 5.

## 2   Double-Multiple Streamtube Model

In a DMST model, the rotor is divided into a front (upstream) and rear (downstream) half-cycle. The flow through a rotor of radius $R$ is discretized into a set of adjacent streamtubes, each featuring two actuator plates in tandem, as illustrated in figure 1. The rotor circumference is divided into $2N_{st}$ arcs of equal length, $S_{st} = R\Delta\theta$, where $\Delta\theta = \pi/N_{st}$ and $N_{st}$ is the number of streamtubes. Each streamtube is defined by an azimuth angle $\theta_{st}$ which depicts the middle point on the rotor arc bounded by the streamtube boundaries, $S_{st} = R\left[\theta_{st} - \frac{\Delta\theta}{2}, \theta_{st} + \frac{\Delta\theta}{2}\right]$. Note that the frontal area of each actuator, $A_{st} = \mathrm{d}hR\Delta\theta\sin\theta_{st}$, in which $\mathrm{d}h$ is the length of a blade element in the spanwise direction, is not constant. Finally, an important aspect of DMST modeling is that an upstream disk is subjected to the free stream velocity, i.e. $U_{in,f} = U_\infty$, while a downstream disk is assumed to be subjected to the wake velocity produced by the front disk, i.e., $U_{in,r} = U_{f,w}$.

Using the above simplified flow description, the DMST model is able to provide predictions based on two methodologies: the momentum theory and the aerodynamic load analysis.

### 2.1   Classical Momentum Theory

The momentum theory aspect of conventional BEM models (including the DMST) builds upon the classical Rankine-Froude actuator disc theory (Betz, 1920; Glauert, 1930). We consider a permeable disk which acts as a drag device slowing the free-stream velocity from $U_\infty$ far upstream, to $U_a$ at the disc plane, and to $U_w$ in the wake. $U_a$ is referred to as the induced velocity and can be expressed in terms of the induction factor $a$ defined as

$$a = 1 - \frac{U_a}{U_\infty}. \tag{1}$$





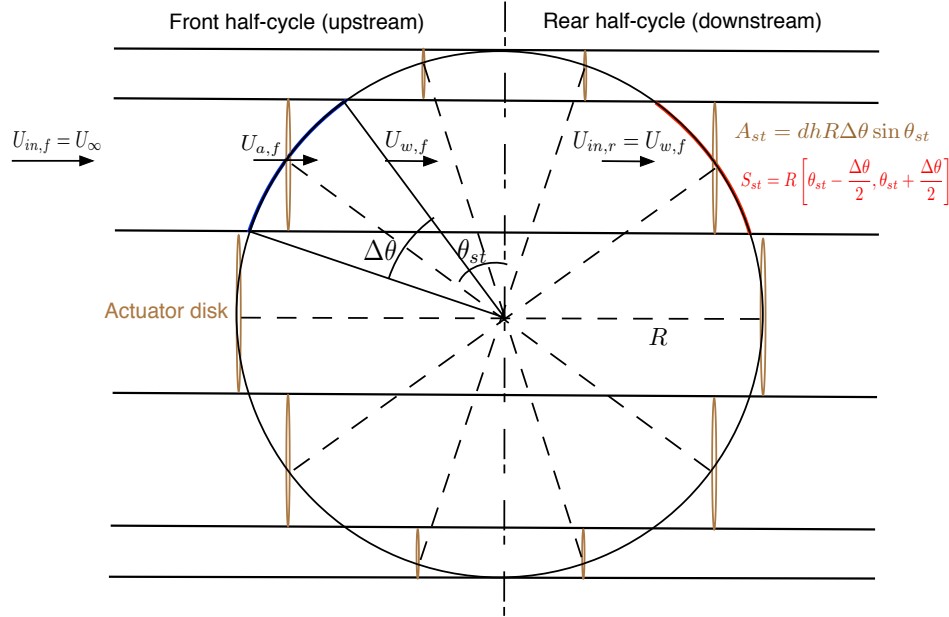

**Figure 1.** Schematic diagram of DMST geometrical configuration with $N_{st} = 5$.

The actuator disk theory assumes potential flow everywhere apart from the immediate vicinity of the disc, a non-rotating actuator disc and no base suction in the wake. The latter assumption implies that the wake can be treated as potential up to a point where the pressure becomes equal to the free-stream pressure, i.e. the boundary condition of the wake becomes $p_w = p_\infty$.

By applying mass and momentum balance to a control volume enclosing the actuator disk and normalizing the resulting drag
with the term $\frac{1}{2}\rho A U_\infty^2$, where $A$ and $\rho$ are the disc area and fluid density, respectively, we obtain the well-known expression for the disk drag coefficient $C_D = 4a(1-a)$ (Hansen, 2015). However, this expression has been shown to agree well with experimental data only for low induction factors (see figure 2). In practice, the following expression is used

$$C_D = \begin{cases} 4a(1-a), & a \leq 0.4 \\ 0.889 - \left(\frac{0.0203-(a-0.143)^2}{0.6427}\right), & 0.4 < a \leq 1 \end{cases} \qquad (2)$$

where the theoretical prediction is applied only for $a < 0.4$, while for larger induction factors Glauert's empirical correction
(Buhl and Marshall, 2005) is used. The wake velocity $U_w(a)$ takes the form

$$U_w = U_\infty(1 - 2a). \qquad (3)$$

Note that for $a > 0.5$ the momentum theory breaks down and predicts negative $U_w$ values. The failure of the theory is even more evident if we inspect the normalized wake cross sectional area, predicted to be $A_w/A = \frac{1-a}{1-2a}$. For $a > 0.5$ this expression yields non-physical negative areas.



Equations (1), (2) and (3) form the basis of the momentum theory which is incorporated in conventional BEM models (including the DMST model).

## 2.2    Current Momentum theory

Steiros and Hultmark (2018) extended the momentum theory of Rankine-Froude by including the effect of base suction in the wake. This theory is based on potential flow principles, where the plate is represented as a distribution of sources of equal
strength. The wake velocities are rescaled, to ensure mass continuity across the plate, while the wake pressure is allowed to assume arbitrary values, so that base suction is taken into account. The various unknown quantities of the problem are then calculated using mass, momentum and energy balances.

The drag coefficient is predicted to be

$$C_D = \frac{4}{3} a \frac{(3-a)}{(1+a)}, \tag{4}$$

which, as shown figure 2, agrees well with experimental data for a larger range of plate porosities, compared to the Rankine-Froude theory. Note that in order to express the drag coefficient as a function of the plate porosity in figure 2, a methodology described in the work of Steiros and Hultmark (2018) is used.

Figure 2 shows that for low plate porosities (less than 20% of the gross plate area) the model of Steiros and Hultmark underpredicts the drag. This is because at low porosities the wake becomes unsteady and vortices are shed from the plate, a
phenomenon which is not modeled by this momentum theory. However, if the wake is stabilized (e.g. with the use of a splitter plate), drag measurements collapse with the theoretical curve for all plate porosities, even up to the solid case (see figure 2).

If we express $C_D$ as a function of the induction factor, we find that the drag predictions of Steiros and Hultmark (2018) agree well with experimental data for $a \leq 0.7$, while a correction is still needed for higher induction factors, to take into account the effect of the unsteadiness of the wake on the drag. Similarly to the classic BEM formulation, we use Glauert's empirical
correction for $a > 0.7$. The resulting drag coefficient is

$$C_D = \begin{cases} \frac{4}{3} a \frac{(3-a)}{(1+a)}, & a \leq 0.7 \\ 0.889 - \left( \frac{0.0203 - (a-0.143)^2}{0.6427} \right), & 0.7 < a \leq 1 \end{cases} \tag{5}$$

which is shown to agree with experimental measurements for all plate porosities (see figure 2). The wake velocity is predicted to be

$$U_w = \frac{1-a}{1+a} U_\infty \tag{6}$$

while the normalized wake width is predicted to be $A_w/A = 1 + a$. Both wake quantities do not assume non-physical infinite or negative values at any induction factor, a fact which further demonstrates that this theory is more suitable than the Rankine-Froude theory for cases of high loading.





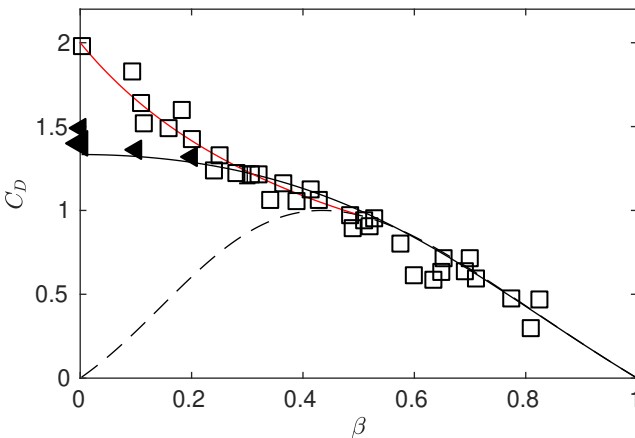

**Figure 2.** Porous plate drag coefficient versus plate open area ratio, $\beta = A_p/A$, where $A_p$ is the porous plate area and $A$ the gross plate area. Squares: measurements with no splitter plate. Triangles: measurements with splitter plate. Solid line: analytical prediction of Steiros and Hultmark (2018). Dashed line: analytical prediction of Rankine-Froude theory. Red line: Glauert's empirical correction. Adapted from Steiros and Hultmark (2018).

Equations 5 and 6 are used instead of equations 2 and 3 in the formulation of the novel DMST algorithm.

## 2.3 Aerodynamic loads analysis

The other aspect of the BEM method deals with the local aerodynamics of a blade segment (airfoil). Figure 3 provides a top-down view of a VAWT rotor and displays a velocity and force diagram associated with a blade section. The blade forces depend on the constantly changing effective angle of attack $\alpha$, which is a function of the azimuth angle, $\theta$, induction factor $a$ and tip speed ratio $\lambda = \frac{\omega R}{U_{in}}$, where $\omega$ is the angular velocity of the turbine and $U_{in} = U_\infty$ for the front streamtubes, while $U_{in} = U_w$ for the rear streamtbues. From the velocity triangle, it can be seen that

$$\alpha = \arctan\left(\frac{(1-a)\sin\theta}{(1-a)\cos\theta + \lambda}\right). \tag{7}$$

The relative velocity experienced by the blade, $U_r$, is the vector sum of the orbital velocity, $\omega R \mathbf{i}_\theta$, and the induced velocity, $U_a \mathbf{i_x}$. By virtue of equation 1 we obtain

$$U_r = U_{in}\left[(1-a)^2 + 2(1-a)\lambda\cos\theta + \lambda^2\right]^{\frac{1}{2}}. \tag{8}$$

Given the angle of attack and relative velocity, aerodynamic loads can be determined using tabulated lift and drag coefficients ($C_L, C_D$) and geometric considerations. In this study, static lift and drag coefficients for the airfoil profile NACA-0021 were collected from the Sandia National Laboratories technical report of Sheldahl and Klimas (1981) for "static"





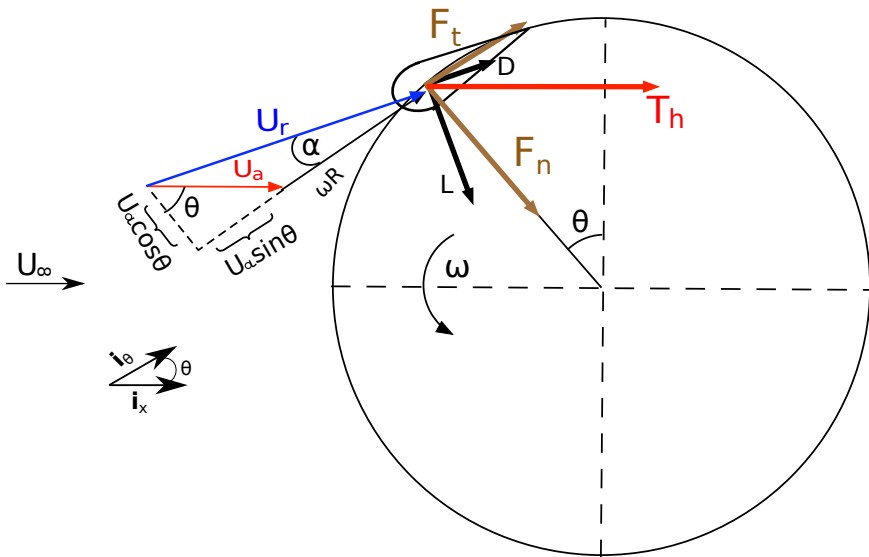

**Figure 3.** Velocity and force diagram on a top-down view of a VAWT rotor.

$Re_{c_N} = [0.36, 0.70, 1.0, 2.0, 5.0] \times 10^6$ and angles of attack $\alpha \in [0°, 180°]$. These static Reynolds numbers are based on the free stream velocity and blade chord length. In the case of a VAWT, the static Reynolds numbers must match the "effective" blade Reynolds number based on the chord length and relative blade velocity. Subsequently, local drag and lift coefficients are
found by interpolation in the $[Re, \alpha]$-space.

It is noted that Sheldahl and Klimas (1981) did not provide measurements for the high $Re$ quantities that we use in this study (in fact high $Re$ airfoil data are sparse in the literature). Instead, they inferred their high-$Re$ data using an airfoil property synthesized code which extended measurements of thinner NACA airfoil profiles, obtained at moderate Reynolds numbers. The above introduces a degree of error in the DMST predictions. Nevertheless, as shown below in the text, the predictions of
the DMST model are relatively accurate for all tested $Re$ and therefore, the inferred data of Sheldahl and Klimas (1981) can be considered reasonable estimations.

The drag and lift coefficients of the airfoils can be combined to yield the local tangential and normal force coefficients

$$C_t = C_D \cos\alpha - C_L \sin\alpha \,, \tag{9}$$

and

$$C_n = C_D \sin\alpha + C_L \cos\alpha \,. \tag{10}$$





By further combining $C_n$ and $C_t$ and multiplying with the local dynamic force $\frac{1}{2}\rho A_b U_r^2$, where $A_b$ is the blade surface, we obtain the instantaneous thrust

$$T_h = \frac{1}{2}\rho A_b U_r^2 \left(C_t \cos\theta + C_n \sin\theta\right). \tag{11}$$

Finally, the torque $\tau$ is the product of the tangential force and radius, $\tau = F_t R$, since in our case these quantities are always orthogonal to each other, i.e.

$$\tau = \frac{1}{2}\rho A_b U_r^2 R C_t. \tag{12}$$

## 2.4 Solving for the induction factor in a streamtube

The DMST model calculates the induction factor $a$, by equating the drag of an actuator disk associated with a given streamtube to the cycle-average thrust on $N_b$ blades that move along the rotor section $S_{st}$.

The cycle-average thrust coefficient corresponding to $N_b$ blades crossing the $i_{st}$ streamtube can be approximated as

$$C_{th}(\theta_{st}, a_{st}) = \frac{\kappa N_b \frac{1}{2\pi}\int_{\Omega_{st}} T_h(\theta, a)d\theta}{\frac{1}{2}\rho A_{st} U_\infty^2}, \tag{13}$$

where the integration domain is $\Omega_{st} = [\theta_{st} - \frac{\Delta\theta}{2}, \theta_{st} + \frac{\Delta\theta}{2}]$. In the limit of infinite number of streamtubes or $N_{st} \to N_\theta$, the integral in eq. (13) reduces to $\Delta\theta T_h(\theta_{st}, a_{st})$. $\kappa$ is a coefficient relevant to the way blade element theory is applied in a VAWT. There is some ambiguity in the literature regarding the value of $\kappa$, which has taken different values in various streamtube algorithm implementations (e.g. $\kappa=1$ (Freris, 1990), $\kappa=2$ (Strickland, 1975) or $\kappa=4$ (Beri and Yao, 2011)). Our experimental data agree well only with the $\kappa = 4$ version, no matter the momentum theory choice; this value is therefore chosen in the DMST model and remains constant in our comparisons of current and conventional momentum theory approaches.

By equating eq. 13 to the drag of the actuator disk related to each streamtube (eq. 2 for the conventional model and eq. 5 for the new model) we obtain a nonlinear equation on $a_{st}$ which we solve iteratively. This process is repeated twice, once for the upstream and once of the downstream rotor half-cycles.

After the induction factor $a_{st}$ has been determined for each streamtube in both the front and rear half-cycles, the total power coefficient $C_p = C_{p,1} + C_{p,2}$, where indices 1 and 2 indicate front and rear contributions, can be computed using

$$C_p = \sum_{i=1}^{2} \frac{\sum_{st=1}^{N_{st}} \frac{\kappa N_b}{2\pi}\int_{\Omega_{st}} \tau_{i,st}(\theta, a_{st})\omega d\theta}{\frac{1}{2}\rho A_d U_\infty^3}, \tag{14}$$

where $A_d$ is the rotor frontal area.



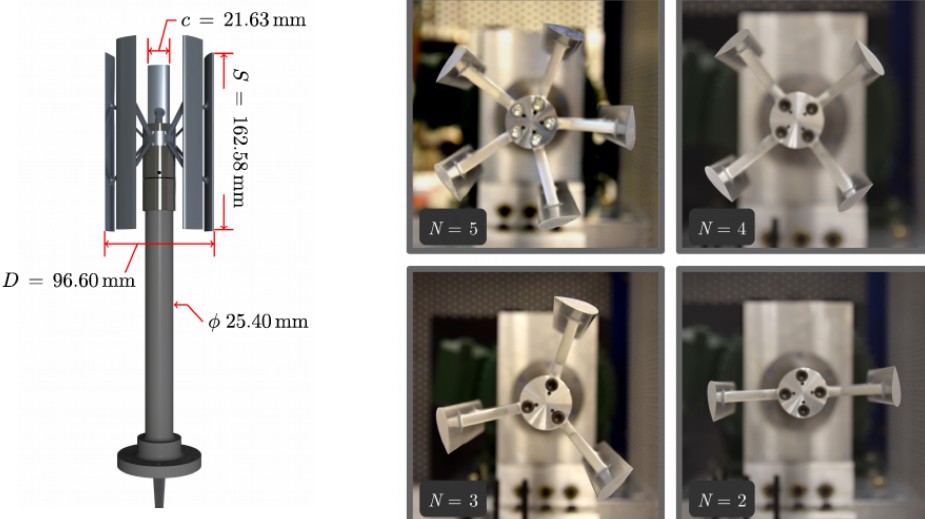

**Figure 4.** Left: Five-bladed VAWT model with dimensions. Right: top-down view of all four turbine configurations. The airfoil profile is that of NACA-0021 in all models.

## 3 Experimental Details

In order to compare the effect of the two momentum theories in the DMST, an experimental campaign of VAWT performace was carried out at Princeton's High Reynolds number Test Facility (HRTF). More details related to this campaign can be found in Duvvuri et al. (2018) and Miller et al. (2018).

The HRTF is a pressurized, low velocity wind tunnel that can be operated at static pressures of up to $p_s = 23$ MPa (230 bar), and free stream velocities of up to $U_\infty = 10$ ms$^{-1}$. This permits the testing of a large range of Reynolds numbers, while keeping the free stream velocities and Mach numbers small. This is a convenient trait which facilitates dynamic similarity between lab and large-scale flows. In our case, the requirement of dynamic similarity was satisfied by simultaneously matching the Reynolds number, tip-speed ratio and Mach number of the VAWT lab-scale models, with those encountered in full scale VAWTs.

A total of four lab-scale VAWT models were tested (see fig. 4), each characterized by its number of blades ($N_b = [2, 3, 4, 5]$). The experiments covered a range of Reynolds numbers ($5.0 \times 10^5 < Re_D < 5 \times 10^6$) and tip-speed-ratios ($0.75 < \lambda < 2.5$). Except from the number of blades, all other turbine features were identical in all four VAWT models (see fig. 4). The airfoil profile was that of a NACA-0021. The rotor was designed to be similar to the one used in the Field Laboratory for Optimized Wind Energy (FLOWE) (Dabiri, 2011).

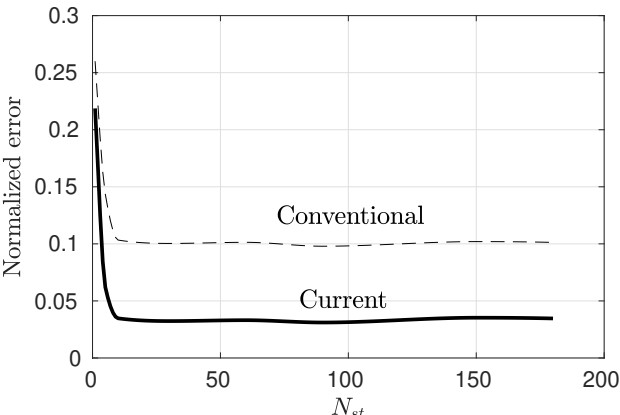

**Figure 5.** Normalized error as a function of number of streamtube shown for both conventional and current DMST models, for a three-bladed turbine at $Re_D = 2.85 \times 10^6$.

## 4 Results and discussion

### 4.1 DMST convergence

In a DMST algorithm, the number of streamtubes, $N_{st}$, is an arbitrary parameter. To decide on that number, a convergence test was performed, based on the "normalized error"

$$\epsilon = \frac{\sum_i \sqrt{(Cp_i - \tilde{C}p_i)^2}}{\sum_i \sqrt{(Cp_i^2 + \tilde{C}p_i^2)}}, \tag{15}$$

where $Cp_i$ and $\tilde{C}p_i$ are the measured and predicted power coefficients for a given tip speed ratio, as indicated by the subscript $i$. In figure 5 we show a typical convergence plot of $\epsilon$. The results are independent of $N_{st}$ after approximately 15 streamtubes. We therefore used for all our tests $N_{st} = 30$. This yielded an average run time of about 0.7 seconds per $\lambda$ case, using a 3.1 GHz Intel Core i7 laptop computer running an in-house Matlab code.

From figure 5 it can be qualitatively seen that the the current DMST model yields more accurate results than the conventional one. In order to assess this increase in accuracy more thoroughly, in the following sections we compare the predicted power coefficients of the two DMST versions for all four turbine configurations and across a range of operating conditions.

### 4.2 Experimental validation

Figure 6 shows predicted and measured power coefficients for a three-bladed VAWT. Results from the current and conventional DMST models are shown on the left and right-hand sides, respectively. The predictions include the total power coefficient $C_p$,





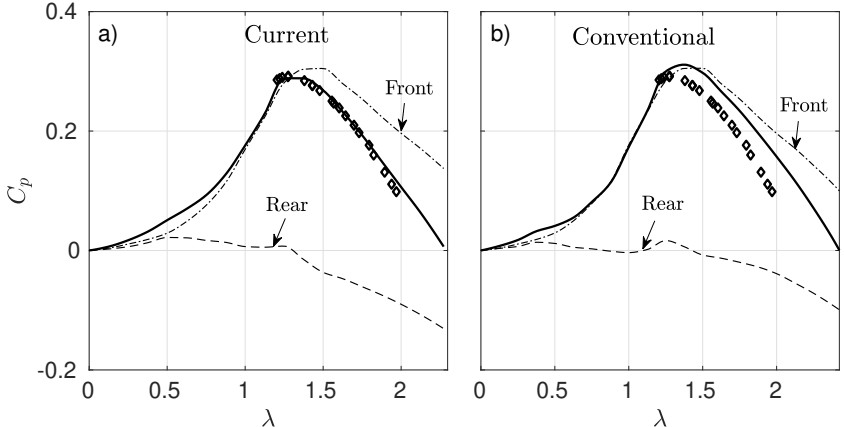

**Figure 6.** Comparison of current (left) and conventional (right) model predictions of power coefficients for a three-bladed VAWT. HRTF experimental data (diamonds) at $Re_D = 2.85 \times 10^6$ are plotted as a reference. Front and rear power contributions $C_{p,f}$ and $C_{p,r}$ are shown explicitly.

and its contributions from the front ($C_{p,f}$) and rear ($C_{p,r}$) disks. The measurements correspond to tip speed ratios $1.20 < \lambda < 1.97$, free stream velocity $U_\infty = 3.1 \text{ ms}^{-1}$ and Reynolds number based on the rotor diameter $Re_D = 2.85 \times 10^6$.

As the tip speed ratio increases, the current DMST model provides power predictions which are in better agreement with the measurements, compared to the conventional one. The reason for this improvement can be seen if we compare the contributions of the front and rear disks for each model. As expected, the front power contributions are very similar, since the input velocity $U_{in,f} = U_\infty$ is the same in both models, and actuator drag is approximately captured by the Glauert correction. However, there is a noticeable difference in the rear half-cycle power predictions, due to the non-negative rear-streamtube input velocities $U_{in,r} = U_{w,f}$ of the new model.

To assess this difference in wake velocity in the above case, in figure 7 we plot the distribution of the upstream wake velocity, $U_{w,f}(\theta)$, at the highest tested tip speed ratio ($\lambda = 1.97$), that is, for the case where base suction (and therefore the difference between the two DMST implementations) is largest. We observe that, indeed, the proposed DMST model, which has the the new momentum theory incorporated, predicts positive wake velocities. The conventional DMST model produces, in general, non-physical negative wake velocities. As seen from the induction factor distribution (right plot in figure 7) the negative wake velocities correspond to $a > 0.5$, in accordance to the Rankine-Froude momentum theory.

For such high induction factors, DMST results based on the Rankine-Froude theory are considered invalid and were not plotted in previous studies (see for instance Ferreira et al. (2014)). Here we show that, even though the new momentum theory certainly improves the predictions by making the DMST model valid for any induction factor, the differences are still small compared to the conventional model. This is because the main source of inaccuracy in the conventional DMST is related to the rear half-rotor, which contributes only a small fraction to the total power coefficient (see figure 6).




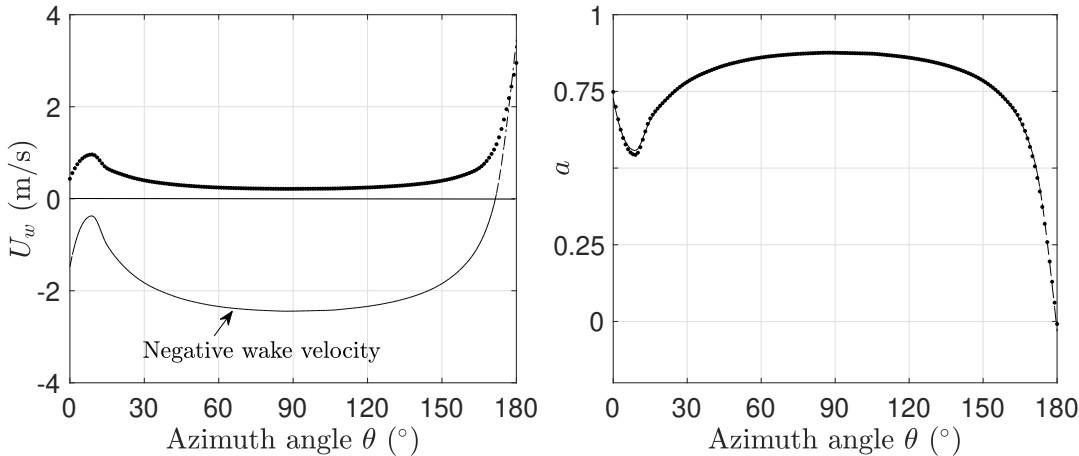

**Figure 7.** Front half-cycle wake velocity profile (left), $U_w$, and induction factor (right), $a$, as a function of azimuth angle $\theta$. Current model (solid line) and conventional model (points) at $\lambda = 1.97$.

Figure 8 shows predicted and measured power coefficients for a three-bladed turbine at four different Reynolds numbers ($Re_D = [1.5, 2.5, 4.5, 6.0] \times 10^6$). In general, the modified DMST agrees quite well with the data and performs consistently better than the conventional DMST model at high tip speed ratios, for all Reynolds numbers. This agreement also suggests that the static airfoil data of Sheldahl and Klimas (1981), which were used in the current DMST implementation, are sufficiently accurate.

In figure 9 we plot the measured and predicted power coefficients for four different VAWT solidities ($N_b =$2, 3, 4 and 5) at constant wind-tunnel conditions ($Re_D = 1.95 \times 10^6$). Again, the proposed DMST formulation improves the predictions as tip speed ratio increases, for all rotor configurations.

## 5 Concluding remarks

A Double-Multiple Streamtube (DMST) model for vertical axis wind turbines (VAWT) is presented, where the classical Rankine-Froude momentum theory is replaced with the momentum theory of Steiros and Hultmark (2018). The classical momentum theory becomes invalid at moderate to high induction factors, and therefore limits the applicability of conventional DMST to small rotor solidities and tip speed ratios, that is, to small rotor loadings. In contrast, the new model introduced here is valid for any induction factor, and therefore renders the DMST applicable, in principle, to any rotor loading.

The predictions of the two DMST formulations were compared with VAWT measurements acquired at Princeton's HRTF facility, covering a range of rotor solidities, tip speed ratios and Reynolds numbers. The data represent both lightly and heavily loaded rotors, in dynamically similar conditions to field-scale VAWTs. The results showed that the new momentum theory improves the predictions of the DMST, especially as tip speed ratio increases. It was found that this improvement is explained

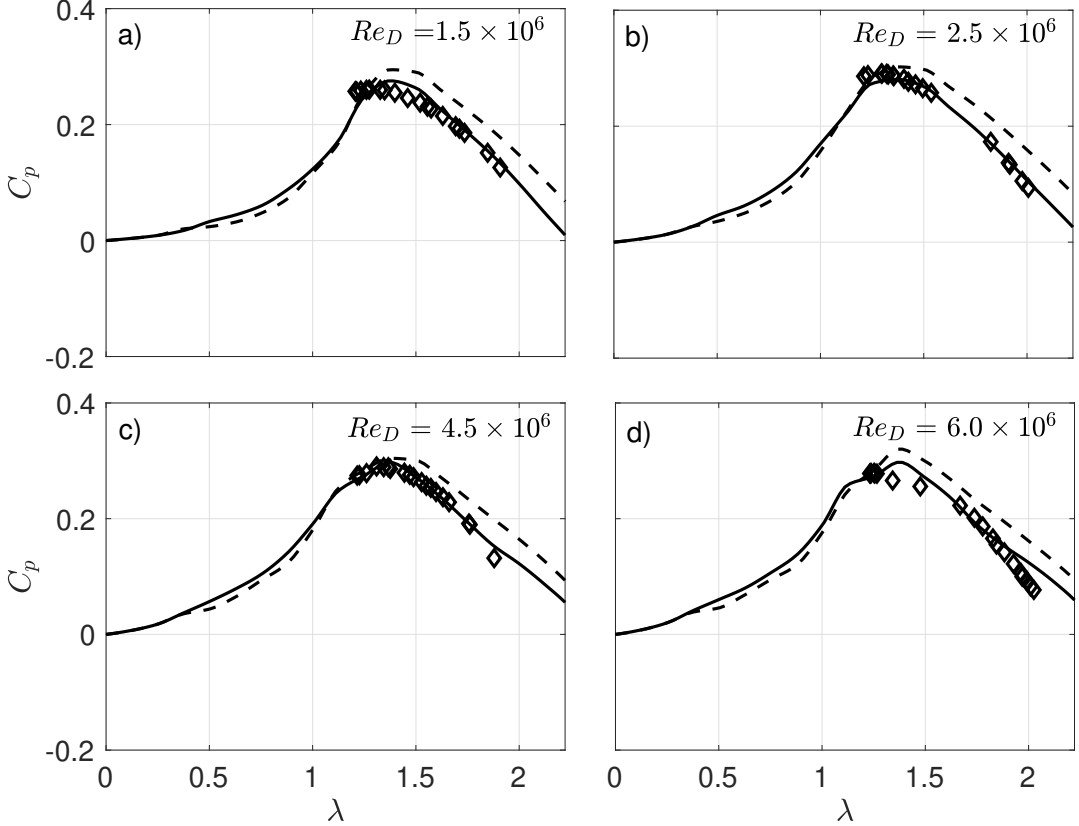

**Figure 8.** Measured (symbols) and predicted (current model: solid lines and conventional model: dashed lines) power coefficients, $C_p$, for a three-bladed VAWT ($N_b = 3$) at $Re_D = [1.5, 2.5, 4.5, 5.0] \times 10^6$.

by a more realistic representation of the wake velocities, or equivalently input velocities to the second rear part of the rotor, from the new momentum theory.

Despite its simplicity and lack of certain flow physics, the DMST model proved reliable in its predictions of the mean power
coefficient of the VAWT, for the tested range of parameters. This could be in part due to the fact that our tested tip speed ratios are rather low, while DMST inaccuracies tend to emerge at high tip speed ratios where friction and wake effects are more significant (Delafin et al., 2017).



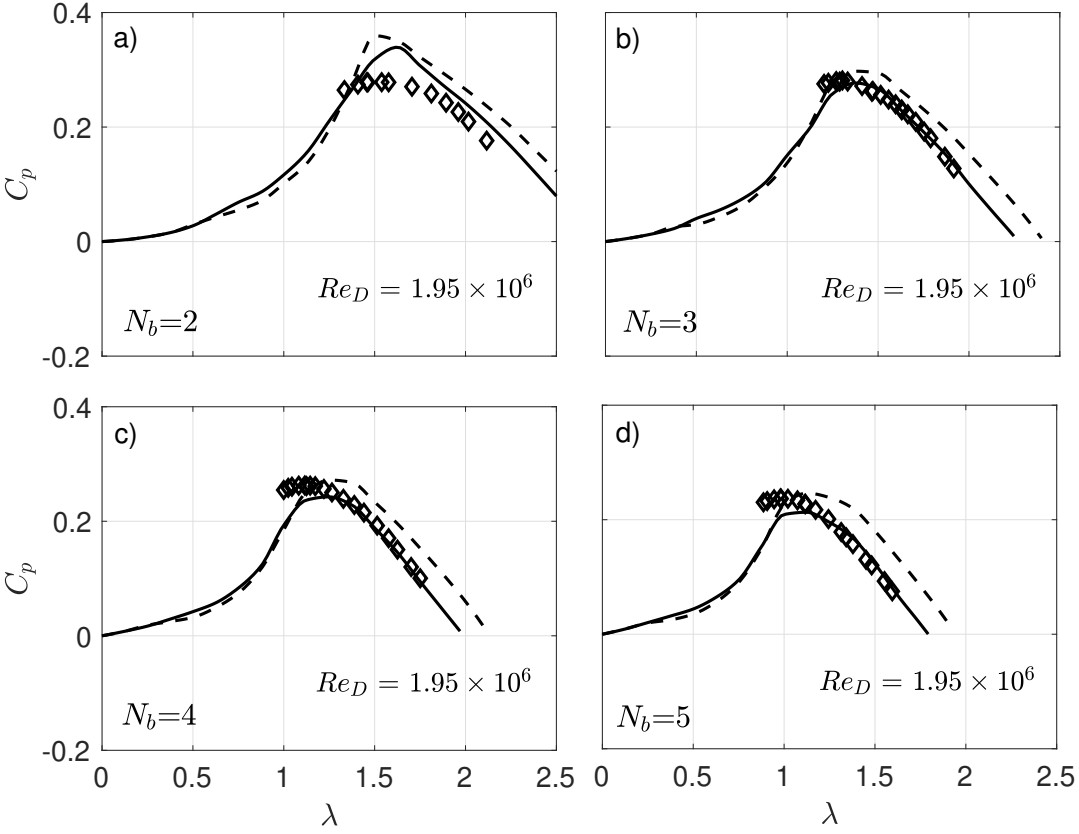

**Figure 9.** Measured (symbols) and predicted (current model: solid lines and conventional model: dashed lines) power coefficients, $C_p$, for four VAWT configurations $N_b = 2, 3, 4, 5$ and at constant $Re_D = 1.95 \times 10^6$.

*Data availability.* Data can be provided upon request. Please contact A. A. Ayati (ayati.anis@gmail.com).

*Competing interests.* The authors declare that they have no conflict of interest.

*Acknowledgements.* The support of the National Science Foundation under grant CBET-1652583 (Program Manager Ron Joslin) is gratefully acknowledged. A. A. Ayati gratefully acknowledges the support of the Akademia-program at the Faculty of Mathematics and Natural Sciences, University of Oslo.



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
