# Peer review of "A Double Multiple Streamtube model for Vertical Axis Wind Turbines of arbitrary rotor loading"

_Wind Energy Science, 2019_

## Referee Comment (RC1) · Hubert BRANGER (Referee) · 22 Sep 2019

Here are some comments on "A Double Multiple Streamtube model for Vertical Axis Wind Turbines of arbitrary rotor loading Âż, by Anis A. Ayati, Konstantinos Steiros, Mark A. Miller, Subrahmanyam Duvvuri and Marcus Hultmark.

The double multiple stream tube model (DMST) is a worldwide analytic model used to predict the flow around vertical axis wind turbines (VAWT). DMST approach is less accurate than CFD codes, but it is far more rapid, far easier to be implemented and far more robust. The authors propose here an improvement, which allows the model to be applicable even to high loaded VAWT, i.e. with high solidity ratios and high tip speed ratios. A new momentum theory is applied to the DMST scheme. This DMST

improvement is tested over direct power measurements on VAWT in the HRTF Princeton Facility. The authors modified the classical Rankine-Froude momentum theory with the new Steiros-Hultmark momentum theory, introducing the base suction effect in the wake. With this trick, the flow past the first half circle may be predicted even if the induction factor "a" is larger than 50% . (normally Uwake= Uinput(1-2a), so if a is greater than $\frac{1}{2}$, Uwake is null or negative . . .).

This new model was tested and compared to experiments performed on a very small VAWT (Radius 4.8 cm , chord length 2.1 cm, blade span: 16 cm !) but with high Reynolds numbers thanks to high density working fluids, and high solidity ratios. The new proposed DMST model provides much better power predictions than the conventional Rankine-Froude model.

It was a pleasure to read this paper. I did not see any mistake. I have just one remark about Figure 7 which seems doubtful: - Figure 7 left : the solid line is supposed to be the current model (see legend) : so why it shows negative velocity values everywhere ? I thought that only the conventional model show negatives values, not the current model: I think there is a mismatch in the figure 7 legend.

- Figure 7 right : why there is absolutely no difference between the solid line (current model) and the conventional one (dot line) apart at teta=10° ?? Induction factors are the same for both models except at 10° ?

However, I have to recall that static lift and drag curves have been used here, and it is known that dynamic stall plays a relevant role in the VAWT problem. Obviously there errors introduced by this fact, whatever the DMST method used. Moreover dynamic stall may be important at low TSR (i.e. in this paper) with hysteresis behavior. In the past, a lot of effort has been invested into developing modifications to the original DMST model to include those effects (Paraschivoiu 2002, Paraschivoiu and Major 1992). Most of the dynamic stall models applied to DMST consist of a series of semi-empirical procedures applied in the calculation of the lift and drag coefficients of the VAWT blade.

So, I wonder what could be the performance of this new DMST model, in comparison with old fashioned DMST model but with dynamic stall corrections.

I personally think that the paper can be published as it is, but a precise check of figure 7 and its legend is required, and a few sentences on DMST models with stall corrections could help the reading.

Paraschivoiu I., "Wind Turbine Design With Emphasis on Darrieus Concept". ISBN 2-553-00931-3. Polytechnic International Press, Montreal, Canada, 2002. Ecole Polytechnique de Montreal.

Paraschivoiu I. and Major S.R., "Indicial Method calculating Dynamic Stall on a Vertical-Axis Wind Turbine.", Journal of Propulsion and Power, 8(4):909-911, 1992.

---

## Referee Comment (RC2) · Anonymous Referee #2 · 2 Oct 2019

The paper is well written and provides a clear description of the improved formulation of the DMST model. The reviewer has only a few suggestions:

1. In section 2.2 it is not entirely clear how the induction factor is linked to the porosity beta. Equation 4 describes CD as a function of the induction factor, while figure 2 describes CD as a function of beta. It would be helpful if the authors elaborate on this.

2. In figure 3, the angle theta is indicated in the wrong corner of the velocity triangle. Consequently, Uacos(theta) and Uasin(theta) also have to be switched in the figure.

3. In figure 7, similar concerns as the other reviewer.

4. line 220 p11 , there is a double 'the the'.

---

## Referee Comment (RC3) · Anonymous Referee #3 · 6 Oct 2019

This is a manuscript on the double multiple streamtube model for horizontal-axis wind turbines. On it, a series of experiments at the high Reynolds number test facility in Princeton show that the recently model proposed by Steiros and Hutmark performs better than the classical Rankine-Froude momentum theory.

Globally, it is an interesting manuscript and the results presented are sound for the wind energy community. Also, the introduction is very well-written and presents an excellent review of state-of-the-art analytical predictive methodologies. I think the paper is suitable for publication on WES. Nevertheless, I have a few minor remarks:

• The experimental set-up section is too short, and some important information is missing. Even if the authors refer to previous works, at least the total blockage and the turbulence intensity of the incoming flow should be reported. Furthermore, as the first

reviewer commented, the pipe's section is small. Previous works from the group state the blockage ratio in almost $9\%$, that would imply the flow near the turbine is modified. Have the authors used any blockage correction to the incoming velocity? If not, can they quantify how measurements are affected? At least the induction factor $a$ will be modified.

• It is not clear which value of $C_d$ is used for the conventional model on figs 6 to 9. Is it the Glauert empirical correction?

• Although this is not important, it is not stated if the computing time of 0.7 secs on line 200 corresponds to both methods (conventional and current).

• On figure 2, what does the case $\beta \sim 0$ mean?

• At the end of line 130, there is a typo: 'streamtbues'.

---

## Author Comment (AC1) · 6 Oct 2019

Dear Prof Hubert Branger,

Thank you very much for your comments and kind words. Below we answer your points one by one.

Regarding figure 7a, there is indeed a mismatch in its legend: The solid line should correspond to the old model, and the points should correspond to the new model. This will be corrected in the revised version.

Regarding figure 7b, it is true that the two models show identical predictions, but that is actually, as figure 7b concerns the front half of the rotor. In the front half, the two momentum theories are almost identical: their inlet velocity is the free stream, while their

drag predictions are almost identical, since Glauert's correction is taken into account (for induction factor greater than 0.4 for the conventional model and greater than 0.7 for the new model, see figure 2 in the manuscript). Therefore, only very slight differences are expected on the predicted quantities of the front half of the rotor, which are due to the small differences of the drag predictions of the two models (when Glauert's correction is included). The difference of predicted quantities is significant only in the rear half of the rotor, where the inlet velocity is now severely dependent on the momentum theory choice of the front half of the rotor. This will be elaborated upon in the revised version.

With respect to dynamic stall, it is very likely that some inaccuracies of the model at low tip speed ratios (i.e. around lambda=1, see figure 9) could be a product of dynamic stall behavior, which is not taken into account in our model. Experimental investigations are ongoing, at Princeton University, in order to quantify and parameterize the dynamic stall effects the same airfoil used for the VAWT turbine. In future work, we envision these results to be incorporated for improved DMST modeling, but it is outside of the scope of this manuscript. Comments on possible effects of dynamic stall in VAWT, and strategies to model such effects in DMST algorithms, will be included in the revised version of the manuscript and the suggested literature will be added.

Yours Sincerely, The authors

---

## Author Comment (AC2) · 7 Oct 2019

Dear Referee,

We would like to thank you for your suggestions and kind words. Below we answer your points one by one.

1. The method to link the open area ratio, beta, and velocity through the plate is proposed in Taylor 1944 (Aero. Res. Counc. R. and M. no. 2237) and expanded in Steiros and Hultmark 2018 (J. Fluid Mech. 853, R3). In summary, the method models the losses of fluid particles that pass through a pore of the plate, by assuming that all kinetic energy which is due to the acceleration of the fluid particle to enter the pore, is lost due to expansion losses. We will elaborate on that on the revised version.

2. We will correct the corner of theta. Thank you for pointing out this error.

3. Regarding figure 7a, there is indeed a mismatch in its legend: The solid line should correspond to the old model, and the points should correspond to the new model. This will be corrected in the revised version.

Regarding figure 7b, it is true that the two models show identical predictions, but that is something to be expected, as figure 7b concerns the front half of the rotor. In the front half, the two momentum theories are almost identical: their inlet velocity is the free stream, while their drag predictions are almost identical, since Glauert's correction is taken into account (for induction factor greater than 0.4 for the conventional model and greater than 0.7 for the new model, see figure 2 in the manuscript). Therefore, only very slight differences are expected on the predicted quantities of the front half of the rotor, which are due to the tiny differences of the drag predictions of the two models (when Glauert's correction is included). The difference of predicted quantities is significant only in the rear half of the rotor, where the inlet velocity is now severely dependent on the momentum theory choice of the front half of the rotor. This will be elaborated in the revised version.

4. The syntax error will be corrected.

Yours Sincerely,

The authors

---

## Author Comment (AC3) · 16 Oct 2019

Dear Referee,

Thank you for your comments and kind words. Below, we answer your points one by one.

We appreciate the reviewer comments regarding the experimental setup section, we agree that additional details would benefit the quality of the paper. To this end, we have included the Rotor Diameter, Span, Chord, Blockage Ratio and inlet turbulence level of the wind tunnel test section in the text.

Regarding the tunnel blockage, the reviewer is correct that the presence of the model will affect the apparent free-stream velocity (and hence performance) of the turbine.
This will be commented in the text. In an effort to mitigate the blockage effect, the authors elected to construct a relatively small model so as to achieve the lowest blockage ratio which is mechanically feasible. No further corrections have been made to the measured performance of the model primarily because the classical correction of Glauert [1], often used for wind tunnel model blockage, assumes quasi-one dimensional flow through the rotor which is only valid at low induction factors a<0.4, when using this type of theory for Blade Element Momentum [2]. As evident in figure 7 of this work, the typical high-solidity VAWT operates far outside the applicability of this basic correction with values exceeding a=0.75. In the future, the authors would like to explore the possibility of creating a new blockage correction specifically aimed at high solidity VAWT operation.

"It is not clear which value of Cd is used for the conventional model on figs 6 to 9. Is it the Glauert empirical correction?"

The drag coefficient formula used for the conventional model is given in equation (2), p. 4 in the manuscript: it is the theoretical prediction of the Rankine-Froude theory for induction factors less than 0.4 and Glauert's empirical correction for larger induction factors. This will be emphasized in the revised version.

"Although this is not important, it is not stated if the computing time of 0.7 secs on line 200 corresponds to both methods (conventional and current)."

The computing time was essentially the same for both models. This will be clarified in the revised version.

"On figure 2, what does the case $\beta \sim 0$ mean?"

This is the case of a solid plate, without any porous area.

"At the end of line 130, there is a typo: 'streamtbues'." The typo will be corrected in the revised version.

Sincerely, The authors.

[1] H. Glauert, Airplane Propellers, in Aerodynamic Theory (Springer, Berlin, Germany, 1935) pp. 169-360.

[2] M.O.L. Hansen, Aerodynamics of Wind Turbines, Vol. 17 (Earthscan, 2007).

---

## Author Response (AR1)

Dear Associate Editor,

Please find a revised version of our manuscript "A Double Multiple Streamtube model for Vertical Axis Wind Turbines of arbitrary rotor loading", with changes in red, as well as detailed responses to the reviewers. We have adopted most of the reviewers' comments in the revised manuscript. The changes include corrections in two figures, additional references and various clarifications.

Reviewer 1: Prof. Hubert Branger

We would like to thank Prof. Branger for the comments and favorable review of the manuscript. Below you will find our responses to the comments which have all been incorporated and helped improve the manuscript.

Here are some comments on A Double Multiple Streamtube model for Vertical Axis Wind Turbines of arbitrary rotor loading Az, by Anis A. Ayati, Konstantinos Steiros, Mark A. Miller, Subrahmanyam Duvvuri and Marcus Hultmark.

The double multiple stream tube model (DMST) is a worldwide analytic model used to predict the flow around vertical axis wind turbines (VAWT). DMST approach is less accurate than CFD codes, but it is far more rapid, far easier to be implemented and far more robust. The authors propose here an improvement, which allows the model to be applicable even to high loaded VAWT, i.e. with high solidity ratios and high tip speed ratios. A new momentum theory is applied to the DMST scheme. This DMST improvement is tested over direct power measurements on VAWT in the HRTF Princeton Facility. The authors modified the classical Rankine-Froude momentum theory with the new Steiros-Hultmark momentum theory, introducing the base suction effect in the wake. With this trick, the flow past the first half circle may be predicted even if the induction factor a is larger than 50% . (normally Uwake= Uinput(1-2a), so if a is greater than $\frac{1}{2}$ , Uwake is null or negative...).

This new model was tested and compared to experiments performed on a very small VAWT (Radius 4.8 cm , chord length 2.1 cm, blade span: 16 cm !) but with high Reynolds numbers thanks to high density working fluids, and high solidity ratios. The new proposed DMST model provides much better power predictions than the conventional Rankine-Froude model.

It was a pleasure to read this paper. I did not see any mistake. I have just one remark about Figure 7 which seems doubtful: - Figure 7 left : the solid line is supposed to be the current model (see legend) : so why it shows negative velocity values everywhere? I thought that only the conventional model show negatives values, not the current model: I think there is a mismatch in the figure 7 legend.

Regarding figure 7a, there is indeed a mismatch in its legend: The solid line should correspond to the old model, and the points should correspond to the new model. This mistake has been corrected in the revised manuscript.

Regarding figure 7b, it is true that the two models show identical predictions, but that is expected, as figure 7b concerns the front half of the rotor. In the front half, the two momentum theories are almost identical (although one is empirical and one analytical): their inlet velocity is the free stream, while their drag predictions are almost identical, since Glauerts correction is taken into account (for induction factor greater than 0.4 for the conventional model and greater than 0.7 for the new model, see figure 2 in the manuscript). Therefore, only very slight differences are expected on the predicted quantities of the front half of the rotor, which are due to the small differences of the drag predictions of the two models (when Glauerts correction is included). The difference of predicted quantities is significant only in the rear half of the rotor, where the inlet velocity is now heavily dependent on the momentum theory choice of the front half of the rotor. This was elaborated, by adding the phrase "The induction factor distribution is almost identical for both DMST formulations, as it concerns the front half of the rotor." in the caption of figure 7.

However, I have to recall that static lift and drag curves have been used here, and it is known that dynamic stall plays a relevant role in the VAWT problem. Obviously there errors introduced by this fact, whatever the DMST method used. Moreover dynamic stall may be important at low TSR (i.e. in this paper) with hysteresis behavior. In the past, a lot of effort has been invested into developing modifications to the original DMST model to include those effects (Paraschivoiu

2002, Paraschivoiu and Major 1992). Most of the dynamic stall models applied to DMST consist of a series of semi-empirical procedures applied in the calculation of the lift and drag coefficients of the VAWT blade. So, I wonder what could be the performance of this new DMST model, in comparison with old fashioned DMST model but with dynamic stall corrections. I personally think that the paper can be published as it is, but a precise check of figure 7 and its legend is required, and a few sentences on DMST models with stall corrections could help the reading.

Paraschivoiu I., Wind Turbine Design With Emphasis on Darrieus Concept. ISBN 2-553-00931-3. Polytechnic International Press, Montreal, Canada, 2002. Ecole Polytechnique de Montreal. Paraschivoiu I. and Major S.R., Indicial Method calculating Dynamic Stall on a Vertical- Axis Wind Turbine., Journal of Propulsion and Power, 8(4):909-911, 1992.

It is very likely that some inaccuracies of the DMST model at low tip speed ratios (i.e. around $\lambda = 1$, see figure 9) could be a product of dynamic stall behavior, which is not taken into account in our model. Experimental investigations are ongoing, at Princeton University, in order to quantify and parameterize the dynamic stall effects on the same airfoil used for the VAWT turbine. In future work, we envision these results to be incorporated for improved DMST modeling, but it is outside of the scope of this manuscript. We have pointed out the possible effect of dynamic stall, by adding the text: "It is noteworthy to mention, however, that static data do not include the effect of dynamic stall, which is an important feature of VAWT, especially at low tip speed ratios. A better agreement of the models can be therefore expected if semiempirical corrections for dynamic stall are included in the DMST algorithm (Paraschivoiu, 2002; Major and Paraschivoiu, 1992).", in line 233 of the manuscript, and we have also added the two references that Prof. Branger suggested.

Reviewer 2
We would like to thank the reviewer for the comments and kind words. Below you will find our reponses to the comments which have all been incorporated in the revised manuscript.

The paper is well written and provides a clear description of the improved formulation of the DMST model. The reviewer has only a few suggestions:
1. In section 2.2 it is not entirely clear how the induction factor is linked to the porosity beta. Equation 4 describes CD as a function of the induction factor, while figure 2 describes CD as a function of beta. It would be helpful if the authors elaborate on this.

The method to link the open area ratio, $\beta$, and velocity through the plate is introduced in Taylor 1944 (Aero. Res. Counc. R. and M. no. 2237) and expanded in Steiros and Hultmark 2018 (J. Fluid Mech. 853, R3). In summary, the method models the losses of fluid particles that pass through a pore of the plate, by assuming that all kinetic energy which is due to the acceleration of the fluid particle to enter the pore, is lost due to expansion losses. This was clarified by adding the text: "Note that in figure 2 the drag coefficient is plotted as a function of the plate porosity, $\beta$, rather than the induction factor $a$. These two quantities can be linked using a methodology described in the work of Steiros and Hultmark (2018), which is based on the modeling of the expansion losses of the fluid that passes through the plate" in line 112 of the manuscript.

2. In figure 3, the angle theta is indicated in the wrong corner of the velocity triangle. Consequently, Uacos(theta) and Uasin(theta) also have to be switched in the figure.

Figure 3 was corrected.

3. In figure 7, similar concerns as the other reviewer.

Regarding figure 7a, there is indeed a mismatch in its legend: The solid line should correspond to the old model, and the points should correspond to the new model. This mistake has been corrected.

Regarding figure 7b, it is true that the two models show identical predictions, but that is expected, as figure 7b concerns the front half of the rotor. In the front half, the two momentum theories are almost identical: their inlet velocity is the free stream, while their drag predictions are almost identical (although one is empirical and one analytical), since Glauerts correction is taken into account (for induction factor greater than 0.4 for the conventional model and greater

than 0.7 for the new model, see figure 2 in the manuscript). Therefore, only very slight differences are expected on the predicted quantities of the front half of the rotor, which are due to the small differences of the drag predictions of the two models (when Glauerts correction is included). The difference of predicted quantities is significant only in the rear half of the rotor, where the inlet velocity is now severely dependent on the momentum theory choice of the front half of the rotor. This was elaborated, by adding the phrase "The induction factor distribution is almost identical for both DMST formulations, as it concerns the front half of the rotor." in the caption of figure 7.

4. line 220 p11 , there is a double the the.

This mistake was corrected.

Reviewer 3
We would like to thank the reviewer for the comments and kind words. Below you will find our reponses to the comments which have all been incorporated and helped improve the manuscript.

This is a manuscript on the double multiple streamtube model for horizontal-axis wind turbines. On it, a series of experiments at the high Reynolds number test facility in Princeton show that the recently model proposed by Steiros and Hutmark performs better than the classical Rankine-Froude momentum theory.

Globally, it is an interesting manuscript and the results presented are sound for the wind energy community. Also, the introduction is very well-written and presents an excellent review of state-of-the-art analytical predictive methodologies. I think the paper is suitable for publication on WES. Nevertheless, I have a few minor remarks:

•The experimental set-up section is too short, and some important information is missing. Even if the authors refer to previous works, at least the total blockage and the turbulence intensity of the incoming flow should be reported. Furthermore, as the first reviewer commented, the pipes section is small. Previous works from the group state the blockage ratio in almost 9%, that would imply the flow near the turbine is modified.

The authors appreciate the reviewer comments regarding the experimental setup section and agree that additional details would benefit the quality of the paper. To this end, section 3 in p. 9 of the manuscript was rewritten, so as to include the basic dimensions of the turbine, inlet turbulence intensities, and wind tunnel blockage due to the presence of the turbine (8.36%).

Have the authors used any blockage correction to the incoming velocity? If not, can they quantify how measurements are affected? At least the induction factor a will be modified.

The reviewer is correct that the presence of the model will affect the apparent free-stream velocity (and hence performance) of the turbine. In an effort to mitigate this, the authors elected to construct a relatively small test-model so as to achieve the lowest blockage ratio which is mechanically feasible. No further corrections have been made to the measured performance of the model primarily because the classical correction of Glauert [1], often used for wind tunnel model blockage assumes quasi-one dimensional flow through the rotor which is only valid at low induction factors $a < 0.4$, when using this type of theory for Blade Element Momentum [2]. As evident in figure 7 of this work, the typical high-solidity VAWT operates far outside the applicability of this basic correction with values exceeding $a = 0.75$. In the future, the authors would like to explore the possibility of creating a new blockage correction specifically aimed at high solidity VAWT operation.

[1] H. Glauert, Airplane Propellers, in Aerodynamic Theory (Springer, Berlin, Germany, 1935) pp. 169-360.

[2] M.O.L. Hansen, Aerodynamics of Wind Turbines, Vol. 17 (Earthscan, 2007).

•It is not clear which value of Cd is used for the conventional model on figs 6 to 9. Is it the Glauert empirical correction?

The drag coefficient used for the conventional model is given by equation (2) in the manuscript. Glauert's correction is used for induction factors greater than 0.4. This is clarified by adding the sentence: "Equations (1), (2) and (3) form the basis of the momentum theory which is incorporated

in conventional BEM models (including the conventional DMST model used in this study)." in line 102 of the manuscript.

•Although this is not important, it is not stated if the computing time of 0.7 secs on line 200 corresponds to both methods (conventional and current).

The computing time was the same in both versions of the DMST model. This was clarified by adding the sentence: "This yielded an average run time of about 0.7 seconds per $\lambda$ case, for both current and conventional DMST algorithms" in line 206 of the manuscript.

•On figure 2, what does the case $\beta \sim 0$ mean?

This is the case of a solid plate, without any porous area.

•At the end of line 130, there is a typo: streamtbues.

The typo was corrected.

[revised manuscript text omitted]